# Parameter and Computation Efficient Transfer Learning for Vision-Language Pre-trained Models

**Qiong Wu**[12], **Wei Yu**[12], **Yiyi Zhou**[12], **Shubin Huang**[1], **Xiaoshuai Sun**[12], **Rongrong Ji**[12*]

[1] Key Laboratory of Multimedia Trusted Perception and Efficient Computing,
Ministry of Education of China, Xiamen University, 361005, P.R. China.
[2] Institute of Artificial Intelligence, Xiamen University, 361005, P.R. China.
`{qiong, weiyu}@stu.xmu.edu.cn, zhouyiyi@xmu.edu.cn,`
`shubinhuang@stu.xmu.edu.cn, {xssun, rrji}@xmu.edu.cn`

## Abstract

With ever increasing parameters and computation, vision-language pre-trained (VLP) models exhibit prohibitive expenditure in downstream task adaption. Recent endeavors mainly focus on parameter efficient transfer learning (PETL) for VLP models by only updating a small number of parameters. However, excessive computational overhead still plagues the application of VLPs. In this paper, we aim at *parameter and computation efficient transfer learning* (PCETL) for VLP models. In particular, PCETL not only needs to limit the number of trainable parameters in VLP models, but also to reduce the computational redundancy during inference, thus enabling a more efficient transfer. To approach this target, we propose a novel *dynamic architecture skipping* (DAS) approach towards PCETL. Instead of directly optimizing the intrinsic architectures of VLP models, DAS first observes the significances of their modules to downstream tasks via a reinforcement learning (RL) based process, and then skips the redundant ones with lightweight networks, *i.e.*, adapters, according to the obtained rewards. In this case, the VLP model can well maintain the scale of trainable parameters while speeding up its inference on downstream tasks. To validate DAS, we apply it to a bunch of representative VLP models, and conduct extensive experiments on a set of VL tasks. The experimental results not only show the great advantages of DAS in reducing computational complexity, *e.g.* $-11.97\%$ FLOPs of METER on VQA2.0, but also confirm its competitiveness against existing PETL methods in terms of parameter scale and performance. Our source code is given in `https://github.com/DoubtedSteam/DAS`.

## 1 Introduction

Inspired by the great success in natural language processing (NLP) [8, 19, 32, 36], large-scale pre-training on massive image-text pairs also becomes the *de-facto* standard in vision-language research [4, 24, 35, 65]. To accommodate the large-scale pre-training corpora, vision-language pre-trained (VLP) models [4, 70, 10, 24, 28, 35, 56, 74] often adopt Transformer-based networks with sheer sizes of parameters and computations. In this case, directly transferring these VLP models to downstream tasks is excessively expensive in terms of memory footprint and computation overhead.

To reduce the costs of pre-training models, recent advances resort to *parameter efficient transfer learning* (PETL) for affordable downstream task adaptions [16, 13, 26, 37, 59, 72, 74, 75]. In particular, the PETL methods aim to save the memory usage for downstream tasks by only updating

---
[*]Corresponding Author.

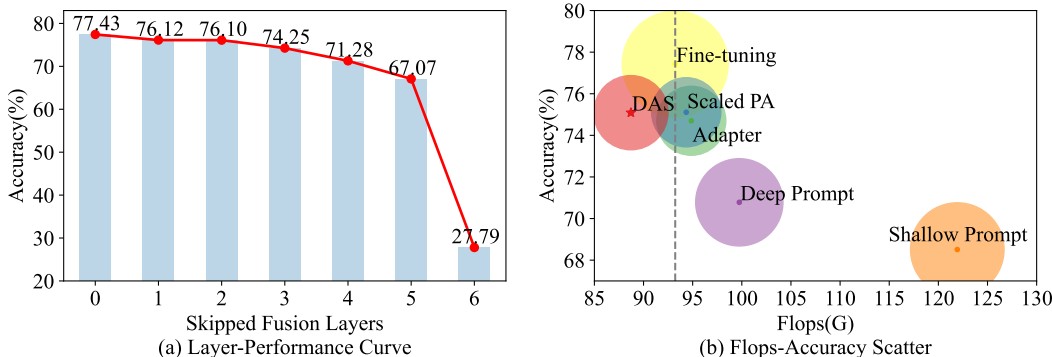

Figure 1: (a) The performance of METER [10] is barely affected when skipping a certain number of its Transformer layers. (b) The comparison on VQA2.0 between the conventional PETL methods [17, 21, 26, 36, 59] and the proposed *Dynamic Architecture Skipping* (DAS) for METER. The circle size represents the memory footprint. DAS is the only method faster than the original VLP model.

or inserting a small number of trainable parameters rather than fully tuning the entire model. For instance, prompt-tuning methods [1, 5, 26, 36, 37, 50, 52, 74, 75] expand the input sequence with hand-craft or learnable tokens to bridge the gap between pre-training and downstream tasks. Practitioners also insert lightweight neural networks called *Adapter* [13, 20, 46, 47, 59, 72, 41] into the pre-trained models, thereby projecting the hidden features onto the semantic space of downstream tasks. More recently, these PETL methods have been successfully introduced to VLP models [13, 33] for either prompt-based image classification [26, 52, 74, 75] or conventional VL tasks like *visual question answering* [58, 59, 43]. Despite the great successes, PETL methods still cannot reduce the computation complexity of VLP models, which is of more significance for applications.

In this paper, we study a novel problem called *parameter and computation efficient transfer learning* (PCETL). To achieve more efficient downstream task adaptions, PCETL is not only expected to maintain the scale of trainable parameters similar to PETL, but more importantly, also needs to reduce the computation complexity of pre-training models, thereby speeding up their inference on downstream tasks. In existing works, the efficiency of the network itself is largely attributed to its manually [6, 22, 44, 54] or automatically structural designs [62, 66, 78, 45]. Although the computation complexity can be further reduced by the compression methods, such as *pruning* [67, 49, 3, 31, 7, 12, 30, 9, 64, 69, 55], *quantification* [11, 29, 73] or *distiiliation* [2, 25], these approaches usually require retraining after optimizing the network architecture, which is not applicable to the VLP models that are well pre-trained on massive data. On one hand, the large-scale pre-training data still needs a certain model capacity to learn these prior knowledge, thus it is hard to obtain a good trade-off between performance and computation overhead for the pre-training objectives. On the other hand, devising a small and effective model for each downstream task is still laborious and expensive, which also contradicts the target of PETL [20, 36], since fully fine-tune is often required.

In this case, we argue that the key to PCETL is to explore the parameter and computation redundancy in existing VLP models. It is generally assumed that the model scale is proportional to the complexity of the task [77, 1, 18, 37, 76]. To robustly serve a variety of downstream tasks, VLP models are pre-trained by multiple pre-training objectives based on tens of millions of image-text pairs [14, 51, 52, 57]. In this case, the excessive parameters are suitable for pre-training, but prone to redundant for a downstream task. As shown in Fig. 1-(a), the performance of METER [10] on VQA is barely affected when skipping a certain number of its Transformer layers. This empirical result also suggests that exploring a short-cut pathway in existing VLP models is a feasible way for PCETL.

To this end, we propose a novel *Dynamic Architecture Skipping* (DAS) approach towards efficient transfer learning for VLP models. By observing the module redundancy of VLP models, DAS can realize the optimal subnetwork routing of VLP models for a downstream task, thereby reducing the computation during inference. In practice, DAS regards this process as a $k$-armed bandit problem, and evaluates the importance of each VL layer/block to the downstream task via numerous subnetwork samplings and quick validations. Thus, the obtained rewards can be used to reflect the redundancy of VL modules and determine which layers to be skipped. Meanwhile, to achieve parameter efficiency,

we also adopt lightweight networks, *i.e.* Adapter [20, 59], to serve the hidden feature adaptions and the short-cut connections of DAS for VLP models.

To validate DAS, we apply it to a set of VLP models, namely including [10], ViLT [28] and LaVIN [42] [2], on three VL benchmarks, namely VQA2.0 [14], NLVR$^2$ [57] and Flickr30K [51]. The experimental results not only show the competitive performance of DAS against the fully finetune and PETL methods [17, 21, 26, 59], but also witness its great advantage in reducing the computation complexity of VLP models. For instance, DAS can help METER achieve $96.60\%$ performance of full tuning on the VQA2.0 benchmark with only $1.65\%$ trainable parameters, while decreasing $11.97\%$ FLOPs. For the practical deployment of a specific VL task, DAS can reduce up to $93.75\%$ parameters of the VLP models [3]. These results well confirm our assumption about the redundancy of VLP models on downstream tasks, and also validated the design of the proposed DAS.

Overall, our contributions can be summarized as three-fold:

- We raise a new problem called *parameter and computation efficient transfer learning* (PCETL) for VLP models, which not only requires to keep the scale of training parameters but also needs to reduce the computation complexity of VLP models on downstream tasks.

- We propose a novel *Dynamic Architecture Skipping* (DAS) for PCETL, which can explore the optimal short-cut pathway in VLP models with the combination of parallel adapters.

- On two VLP models and three benchmark datasets, the proposed DAS not only reduces the computation overhead by a large extent, *e.g.*, $-11.97\%$ FLOPs of METER on VQA2.0, but also is on par with existing PETL methods in terms of parameter and performance.

## 2   Related Work

### 2.1   Vision-Language Pre-training

In recent years, the advancement in natural language processing (NLP) [32, 36] also sparks the prevalence of large-scale pre-training in vision-language (VL) research [4, 10, 24, 28, 35, 56, 74]. In particular, VL pre-training also accomplishes self-supervised learning on massive image-text pairs based on the generative prediction tasks, *e.g. masked language modeling* (MLM) and *masked image modeling* (MIM). Furthermore, *Image Text Matching* (ITM) [10, 28] is also applied to align two modalities. In terms of network architecture, most VLP models are equipped with two modality encoders to extract the visual and text features, *e.g.* BERT [8] and Faster-RCNN [53], respectively, based on which a stack of Transformer-based layers are deployed for cross-modal fusions [4, 10, 23, 24, 28, 34, 38, 39, 56, 61, 63]. For instance, ViL-BERT [39] and LXMERT [61] contain two independent Transformer-based branches [63] for region and text feature extractions, and another Transformer block is used for cross-modal interaction. To simplify the framework, Visual-BERT [34], VL-BERT [56] and UNITER [4] abandon additional branches and directly feed features into a single Transformer network. Then Pixel-BERT [24], CLIP-ViL [38], and METER [10] break the limitation of object detection backbones by directly applying grid features for multi-modal pre-training. To further simplify the model complexity, ViLT [28] directly feeds the word embeddings and image patches into the Transformer blocks. Additionally, CLIP [52] applies cross-modal contrastive learning for vision-language alignment with a shallow fusion layer for prediction. Overall, these VLP models often require more network branches due to the increase of modalities, resulting in more parameter and computation overheads. In this paper, we present the first attempt to evaluate their network redundancy on downstream tasks.

### 2.2   Parameter Efficient Transfer Learning

Parameter Efficient Transfer Learning (PETL) [13, 15, 17, 20, 21, 46–48, 58–60, 71, 72] aims to approach the fully-tuned performance on downstream by updating a small number of parameters. One of the main methodology in PETL is prompt-tuning [1, 5, 26, 36, 37, 50, 52, 74, 75], which is originally designed for large pre-trained language models such as GPT-3 [1]. Concretely, the hand-craft prompts [50, 52] expand the original input sequence with natural language and regard all

---

[2]Due to the page limit, the results of LaVIN are given in our Github project.

[3]When the model is only deployed for a task, the skipped layers can be also removed during deployment.

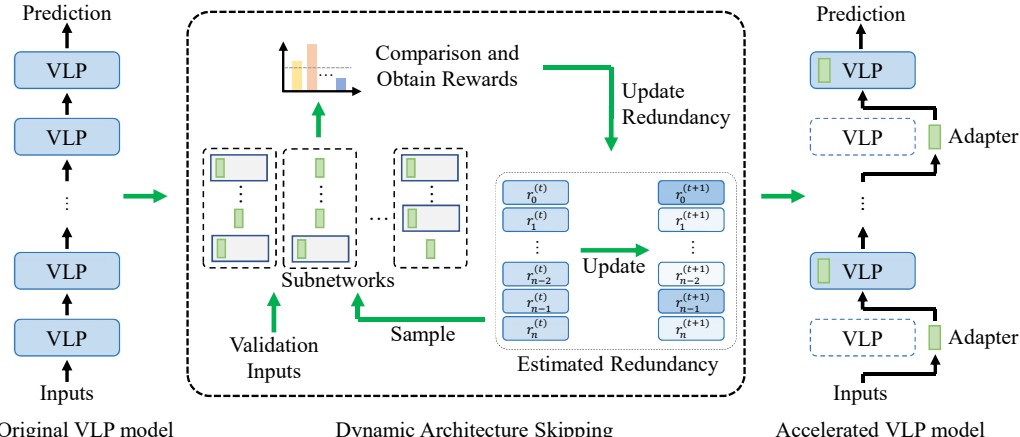

Figure 2: Illustration of *Dynamic Architecture Skipping* (DAS). DAS regards the network skipping as a $k$-armed bandit problem, and evaluates the redundancy of each VL layer/block via numerous subnetwork samplings. The accumulated rewards are used to determine which layers can be skipped, and adapters are also used for feature adaptions and short-cut connections.

problems as a generation task. To better fit downstream tasks, soft prompt tuning methods [26, 36, 75] replace the handcraft prompts with a sequence of trainable embeddings. In addition to prompt-tuning, adapter-based [13, 20, 46, 47, 59, 72] methods insert lightweight feed-forward networks into VLP models, and these methods transfer VLP models by projecting hidden features onto the downstream distributions [20, 59]. Furthermore, LoRA [21] is proposed to transfer VLP models without additional calculation overhead in the inference stage by updating the low-rank parts of the original parameters. Besides, Side-tuning [71] runs in parallel with the pre-trained models to adapt downstream tasks while overcoming the constraint from the concrete structure. In addition, LST [58] stacks the outputs of the pre-trained modules in a parallel path. Without feedback to the VLP model, LST alleviates the memory requirement in the transfer while increasing the computation overhead. Compared to fine-tuning the entire model, PETL methods significantly improve the efficiency in transferring VLP models to downstream tasks. However, all of the above methods take the original VLP model as the upper bound of inference efficiency. In this paper, the proposed DAS method is the first to reduce the computation of VLP models while maintaining competitive performance. In terms of computation efficiency, network compression methods can also reduce the computation overhead ,but they often require to fully tune the model on the downstream tasks, such as LayerDrop [12], EfficientVLM [64] and J2C [9]. This setting make them against the target of PCETL about parameter efficiency.

## 3  Preliminary

We first revisit the principle of PETL methods for VLP models. Given a vision-language pre-trained (VLP) model, denoted as $G(\cdot)$, the target of PETL is to achieve the parameter-efficient adaption on the downstream task, which can be summarized as

$$\underset{\sigma}{\arg\min} \, \mathcal{L}\big(G(I, T | [\boldsymbol{\theta}, \boldsymbol{\sigma}])\big), \tag{1}$$

where $\boldsymbol{\theta} = \{\theta_1, \theta_2, .., \theta_n\}$ represent the parameters of $n$ layers in the VLP model, and $\boldsymbol{\theta}$ is usually frozen in PETL. $(I, T)$ denotes the image-text pair, and $\boldsymbol{\sigma}$ is a small number of updated parameters. Since all VLP layers are reserved on downstream tasks, PETL methods can only reduce the parameter expenditure but not the computation of VLP models. Moreover, most PETL methods often incur non-negligible latency during inference [21, 40].

According to the observation in Fig 1-(a), there exists obvious redundancy in the VLP models. To this end, the objective of the proposed task of *parameter and computation efficient transfer learning* (PCETL) can be defined by

$$\underset{\boldsymbol{\sigma}, \mathbf{K}}{\arg\min} \, \mathcal{L}\big(G(I, T | [\boldsymbol{\theta}_{\mathbf{K}}, \boldsymbol{\sigma}])\big), \tag{2}$$

---

**Algorithm 1** Dynamic Architecture Skipping

---

**Require:** The training and validation sets, $D_t$, $D_v$. VLP modules $\{\theta_i\}$. Adapters $\{\sigma_i\}$.
**Ensure:** The optimal network structure $G_s$.

   Initialize the degree of redundancy of all modules, $\mathbf{r}^{(0)} = [\mathbf{r}_1^{(0)}, \mathbf{r}_2^{(0)}, ..., \mathbf{r}_n^{(0)}]$
   **for** $t$ in $T$ **do**
      Calculate the score according to Eq. 3 and sample a subnetwork.
      Update the parameter of adapters for adaptation and substitute $\sigma$ by $\mathcal{L}_{train}$.
      **if** $t$ mod $interval == 0$ **then**
         Obtain a validation batch $d_v \leftarrow D_v$.
         Evaluate the performance of $c$ subnetworks and get their reward $r_i = e^{-\mathcal{L}_{val}}$
         Update the degree of redundancy $\mathbf{r}^{(t)}$ according to Eq. 5.
      **end if**
   **end for**
   Obtain the optimal structure $\boldsymbol{\theta_K}$ based on $\mathbf{r}$.

---

where $\boldsymbol{\theta_K} = \{\theta_{k_1}, \theta_{k_2}, ..., \theta_{k_m}\} \in \boldsymbol{\theta}$ is the parameters of VLP modules except the skipped ones. Via skipping the redundant layers in VLP models and the combination of PETL methods, VLP models can accelerate the inference speed and maintain the scale of updated parameters.

## 4 Dynamic Architecture Skipping

### 4.1 Redundancy Estimation

In this paper, we propose a novel transfer learning approach called *Dynamic Architecture Skipping* (DAS) towards the parameter and computation efficient adaption of VLP models.

DAS first observes the model redundancy to downstream tasks before skipping the layers of VLP models. In practice, we regard this process as a $k$-armed bandit problem, as illustrated in Fig. 2. Firstly, we define the degree of redundancy as $\mathbf{r} \in \mathbb{R}^n$, where $n$ is the number of VLP modules. To correctly estimate the redundancy, we equally initialize $\mathbf{r}_i = 0$ and update it momentously.

In each training step, we skip $m$ modules according to uniform distribution based on $\mathbf{r}$, and train the sampled architectures on the downstream data. For $t$-th step, the action policy $\pi_i^{(t)}$ for the $i$-th VLP module follows the distribution:

$$\pi_i^{(t)} \sim U(0, \rho(\mathbf{r}_i^{(t)})), \tag{3}$$

where $U(a, b)$ is the uniform distribution between $a$ and $b$. And $\rho$ represent the Sigmoid function. We randomly pick a probability from the $\pi_i^{(t)}$ of each module as the score $s_i^{(t)}$. According to the score $s_i^{(t)}$, the sampled subnetwork can be defined by

$$G_s = g_0 \circ g_1 \circ ... \circ g_n,$$
$$where \quad g_i = \begin{cases} \theta_i, i \in \{j | s_j^{(t)} < s_m^{(t)}\}, \\ \sigma_i, i \in \{j | s_j^{(t)} \geq s_m^{(t)}\}. \end{cases} \tag{4}$$

Here, $g_i \circ g_{i+1}$ represents the compositional function $g_i(g_{i+1}(\cdot))$. $\theta_i$ denotes the original VL module, and $\sigma_i$ is the lightweight module like adapter for short-cut connection. And $s_m^{(t)}$ are the $m$ largest values in the picked scores. Here, the module with a larger $\mathbf{r}_i^{(t)}$ is more likely to be skipped during training. Meanwhile, Eq. 4 also help $\sigma_i$ learn pre-trained knowledge from $\theta_i$ in a distillation way [68].

Then, DAS observes the redundancy of VLP modules in a reinforcement learning manner, as shown in 2. DAS samples $c$ candidate network structures and calculates their rewards according to their loss values during validation, *i.e.* reward $v = e^{-loss}$. Based on the rewards, the degree of redundancy $r$ can be updated by

$$\mathbf{r}_i^{(t+1)} = \mathbf{r}_i^{(t)} + (v_h - \frac{1}{c}\sum_{j=1}^{c} v_j). \tag{5}$$

Here, $v_h$ denotes the reward of the sampled subnetwork, where the $i$-th module is skipped. When its validation loss is larger than the mean value, it suggests that this skipped module is more redundant. Eq. 5 is conducted at short training intervals to makes sure that most subnetworks can be sufficiently validated via numerous samplings, and the theorem of large numbers can guarantee the optimality of search results. The detailed search procedure of DAS is illustrated in Algorithm. 1. Finally, according to the degree of redundancy $\mathbf{r}$, we can select top-$m$ layers to be skipped, thereby reducing the computation complexity of VLP models.

## 4.2   Model Adapting

To reduce the updated parameter scale during adaptation, DAS also introduces lightweight adapters [20, 59] to serve the hidden feature transformations as well as the short-cut connections in VLP models. Typically, an adapter is constructed by two linear projection layers and an activation function in between:

$$adapter(\mathbf{x}) = ReLU(\mathbf{x}\mathbf{W}_{in})\mathbf{W}_{out}. \tag{6}$$

Here, $\mathbf{W}_{in} \in \mathbb{R}^{d \times h}$ and $\mathbf{W}_{out} \in \mathbb{R}^{h \times d}$ are two trainable matrices, where $h \ll d$. For the $i$-th VLP module, the adaptation can be defined by

$$\mathbf{x}_i = \mathbf{x}_{i-1} + VLP(\mathbf{x}_{i-1}) + adapter(\mathbf{x}_{i-1}), \tag{7}$$

where $\mathbf{x}_i$ is the output of the $i$-th component. In this manner, DAS can freeze most parameters in the VLP models during adaption, similar to existing PETL methods [59].

Notably, directly removing the redundant modules will make the subsequent layers to receive the hidden features with drastic changes. Meanwhile, we do not expect the fully tuning of the whole model. In this case, we apply the adapter to serve the short-cut connection of the skipped layers:

$$\mathbf{x}_i = \mathbf{x}_{i-1} + adapter_r(\mathbf{x}_{i-1}). \tag{8}$$

In this way, DAS can not only bridge the gap between feature transformations, but also retain parameter efficiency. Based on the estimated redundancy, DAS skips the redundant modules and finds out the optimal pathway for the downstream task with the helps of adapters, as shown in Fig. 2.

## 5   Experiment

### 5.1   Datasets and Experimental Setup

**Visual Question Answering**. We conduct experiments on VQA2.0 [14]. Instead of answering the question in open-ended natural language, it is converted into a classification task with $3,129$ classes. Following the previous setting [10, 28], the PETL methods and DAS are trained on the train and validation sets of VQA2.0, and we report the *test-dev* results from the online evaluation [4].

**Natural Language for Visual Reasoning**. The NLVR$^2$ [57] is a dataset for classifying triplets of two images and a question into two classes. Because its form is different from the setup of VLP models, which has two images in one VL example, we feed these triplet examples to the model following the default setting of ViLT [28] and METER [10]. Under this setting, the paired images and the question are input to the network, respectively. And the classifier predicts the results according to the concatenation of two representations.

**Retrieval Task**. For cross-modal retrieval, we measure the performance on Flickr30K [51] re-splited by Karpathy *et al.* [27]. We initialize the predictor for similarity measurement from the pre-trained ITM head. During the training, we randomly sample 15 instances as negative samples.

### 5.2   Implementation details

We validate DAS on two deep-fusion based VLP models, which are ViLT [28] and METER [10]. In terms of ViLT, we update the parameters of the additional components, classifier, class token and modal-type embeddings, while the rest are frozen. Following the most conventional setting [17, 59], the width of hidden states in adapters is set to 96. And the hidden dimension of the adapter used for

---

[4]https://eval.ai/web/challenges/challenge-page/830/overview

Table 1: Comparison between DAS and the PETL methods for METER and ViLT on VQA, NLVR$^2$ and Flickr30K. The best performance is **bold** and the second best is underlined.

| | | VQA | | NLVR$^2$ | | Flickr30K | | Avg. | |
|---|---|---|---|---|---|---|---|---|---|
| **METER** | | | | | | | | | |
| **Method** | **Updated Parameter** | **test-dev** | **Additional FLOPs** | **test-P** | **Additional FLOPs** | **IR/TR R@1** | **Additional FLOPs** | **Per.** | **Additional FLOPs** |
| Full Tuning | 323.31M | 77.43 | 0.00 | 83.05 | 0.00 | 82.22/94.30 | 0.00 | 84.25 | 0.00 |
| Classifier Only | - | 69.93 | 0.00 | 73.23 | 0.00 | 78.80/89.00 | 0.00 | 77.74 | 0.00 |
| Shallow Prompt [36] | 0.30M | 68.51 | +28.71G | 65.69 | 26.84G | 74.20/88.60 | +28.71G | 74.25 | +28.71G |
| Deep Prompt [26] | 1.84M | 70.78 | +6.53G | 72.64 | +5.59G | 78.84/89.40 | +6.53G | 77.92 | +6.53G |
| LoRA [21] | 0.29M | 74.00 | 0.00 | 78.82 | 0.00 | 79.86/92.60 | 0.00 | 81.32 | 0.00 |
| Adapter [59] | 5.34M | 74.70 | +1.64G | 79.93 | +1.38G | 80.38/91.90 | +1.64G | 81.73 | +1.64G |
| Scaled PA [17] | 3.59M | **75.11** | +1.12G | 80.38 | +0.66G | 80.40/**93.20** | +1.12G | **82.27** | +1.12G |
| **DAS$_4$-Fusion** | 5.34M | 74.80 | **-11.16G** | 80.11 | **-5.13G** | 80.12/91.80 | **-11.16G** | 81.71 | **-9.15G** |
| **DAS$_4$-Global** | 6.23M | 75.09 | -4.51G | **80.69** | -3.67G | **80.42**/91.40 | -6.06G | 81.90 | -4.74G |
| **ViLT** | | | | | | | | | |
| **Method** | **Updated Parameter** | **test-dev** | **Additional FLOPs** | **test-P** | **Additional FLOPs** | **IR/TR R@1** | **Additional FLOPs** | **Per.** | **Additional FLOPs** |
| Full Tuning | 115.43M | 71.26 | 0.00 | 76.13 | 0.00 | 64.40/83.50 | 0.00 | 73.82 | 0.00 |
| Classifier Only | - | 65.75 | 0.00 | 66.08 | 0.00 | 57.42/78.00 | 0.00 | 66.81 | 0.00 |
| Shallow Prompt [36] | 0.15M | 66.47 | +19.53G | 66.47 | +19.53G | 55.92/74.80 | +19.53G | 65.92 | +19.53G |
| Deep Prompt [26] | 1.84M | 69.30 | +5.14G | 73.34 | +5.14G | 58.64/79.50 | +5.14G | 70.20 | +5.14G |
| LoRA [21] | 0.15M | 68.44 | 0.00 | 72.77 | 0.00 | 57.44/77.70 | 0.00 | 69.09 | 0.00 |
| Scaled PA [17] | 1.80M | 70.40 | +0.44G | 75.13 | +0.44G | 61.88/79.00 | +0.44G | 71.60 | +0.44G |
| Adapter [59] | 3.56M | **70.85** | +0.86G | **75.51** | +0.86G | **62.68/81.40** | +0.86G | **72.61** | +0.86G |
| **DAS$_1$** | 3.56M | 69.28 | **-1.03G** | 74.89 | **-1.03G** | 60.66/80.80 | **-1.03G** | 71.41 | **-1.03G** |

the skip connection is set to 192 to retain a certain capacity. The VLP model is first warmed up for one epoch. In this epoch, the subnetwork is randomly sampled according to the skipped number $m$. Then the search runs 2 epochs and the redundancy observation is executed at 10-th step per interval. Finally, the optimal architecture will be trained for another 10 epochs. Notably, the validation set is used during training for all methods. In terms of METER, we split its fusion layer into two modules, *i.e.* the vision and language ones, which are skipped separately. The hidden dimension of the adapter used as the skip connection is set to 192 for encoders and 288 for fusion layers. The rest settings are the same as ViLT. We conduct all experiments with a single NVIDIA Tesla A100 GPU and the settings not mentioned are the same as ViLT [28] and METER [10].

## 5.3 Experimental results

### 5.3.1 Quantitative analysis

**Comparion with PETL methods.** We first compare DAS with a bunch of PETL methods [17, 21, 26, 36, 59] on the VLP models, of which results are given in Tab. 1. Here, **the suffix of DAS** denotes the number of skipped layers, and "*fusion*" and "*global*" refer to the range of network skipping, *i.e.*, only the fusion branch or the complete model. From Tab. 1, the first observation is that existing PETL methods can largely reduce the amount of updated parameters for VLP models. For instance, the prompt-based methods only require about 1M parameters for two VLP models, nearly 300 times less than full tuning. Meanwhile, their performance gap to fully tuning is also marginal, *e.g.*, Scaled PA [17]. However, we can also see that all these PETL methods cannot reduce the computation of VLP models, and some of them incurs obvious increases in FLOPs, *e.g.* +28.71G by Shallow Prompt [36] on METER. The most computation efficient one is LoRA [21], which applies the re-parameterization technique to merge the additional modules into the VLP model, taking no extra computations. However, its performance obviously lags behind other PETL methods and our DAS. In stark contrast, DAS is the only method that can reduce the computation overhead on the downstream VL tasks, *e.g.*, -11.16G FLOPs by DAS$_4$-Fusion on VQA. More importantly, its updated parameter scale is only slightly larger than Adapter and Scaled PA, while the overall performance is still competitive. These results well confirm the effectiveness of DAS towards PCETL.

**Ablation of the number of skipped layers.** In Tab. 2, we report the results of skipping different numbers of layers by DAS. In terms of METER, we can first observe that skipping a few layers

Table 2: Ablation study of different numbers of skipped layers by DAS. "Fusion" denotes the skipping is only for the fusion branch, while "Global" refers to the entire METER.

| Method | Skipped Number | VQA | | NLVR$^2$ | | Avg. | |
|---|---|---|---|---|---|---|---|
| | | test-dev | Additional FLOPs | test-P | Additional FLOPs | Per. | Additional FLOPs |
| **METER** | | | | | | | |
| Baseline | 0 | 75.28 | 1.68G | 81.28 | 0.99G | 78.28 | 1.34G |
| DAS-Fusion | 2 | 74.92 | -9.06G | 80.07 | -2.66G | 77.50 | -5.86G |
| | 4 | 74.80 | -11.16G | 80.11 | -4.14G | 77.46 | -7.65G |
| | 6 | 74.67 | -17.58G | 78.16 | -9.97G | 76.42 | -13.78G |
| | 8 | 73.70 | -24.00G | 79.30 | -11.45G | 76.50 | -17.72G |
| DAS-Global | 2 | 75.24 | -3.96G | 81.37 | -2.19G | 78.31 | -3.08G |
| | 4 | 75.13 | -4.51G | 81.34 | -3.67G | 78.24 | -4.09G |
| | 6 | 75.02 | -5.06G | 80.04 | -4.22G | 77.53 | -4.64G |
| | 8 | 74.95 | -5.61G | 79.61 | -8.34G | 77.28 | -6.97G |
| **ViLT** | | | | | | | |
| Baseline | 0 | 70.13 | 0.73G | 76.26 | 0.73G | 73.20 | 0.73G |
| DAS | 1 | 69.28 | -1.03G | 74.89 | -1.03G | 72.09 | -1.03G |
| | 2 | 67.64 | -2.79 | 73.00 | -2.79G | 70.32 | -2.79G |

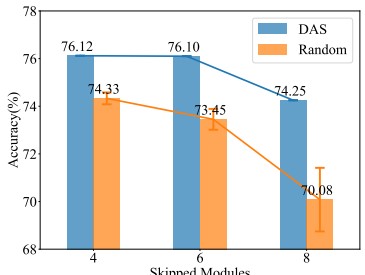

Figure 3: The comparison between DAS and random skipping for METER on the VQA2.0.

Table 3: Computation overhead of different methods for METER on the VQA2.0.

| Method | VQA test-dev | Additional FLOPs | METER Training | | Testing | |
|---|---|---|---|---|---|---|
| | | | Memory(G) | Time(h) | Memory(G) | Speed(Sample/s) |
| Full Tuning | 77.43 | 0 | >40 | N/A | 6.8 | 4.16 |
| LoRA | 74.00 | 0 | 21.5 | 27 | 6.8 | 4.16 (+0.00%) |
| Adapter | 74.70 | +1.64G | 22.9 | 28 | 7.2 | 4.09 (-1.68%) |
| Scaled PA | 75.11 | +1.12G | 23.1 | 30 | 7.1 | 3.95 (-5.04%) |
| DAS$_4$-Global | 75.09 | -4.51G | 21.7 (search) & 20.6 (train) | 10 (search) + 20 (train) | 6.5 | 4.57 (+9.85%) |
| DAS$_4$-Fusion | 74.80 | -11.16G | 21.7 (search) & 18.1 (train) | 10 (search) + 18 (train) | 6.5 | 4.96 (+19.23%) |

has limited impact on its performance, *e.g.*, skipping up to 8 fusion layers only has about 2.2% performance drops, strongly suggesting the redundancy of this VLP model. However, DAS can only reduce about one layer for ViLT without notable performance degradations. To explain, METER is a deep VLP model with two independent encoders to extract the features of image and text, while ViLT processes the multi-modal information from image pixels and word embeddings. In this case, ViLT is more compact than METER for downstream tasks, and this is also reflected in their parameter scales and performance. In terms of METER, we can also see the difference in optimizing the fusion branch and the entire model, *i.e.* DAS-Fusion and DAS-Global. With the same number of skipped layers, DAS-Fusion can reduce more FLOPs since the length of multi-modal inputs is often larger than the single-modal one. Meanwhile, when evaluating the entire model, DAS often tends to reduce the layers in the language encoders [5], which also suggests that natural language understanding is often less difficult in the VL tasks. Overall, these results well confirm our motivation about the redundancy of VLP models in downstream VL tasks, especially the ones with independent encoders like METER.

**Reliability of DAS.** Considering that DAS is an RL-based search approach, we also examine its stability and reliability via comparing to random skipping, of which results are given in Fig. 3. It can be seen that DAS is consistently better than random skipping without regard to the number of skipping layers, well confirming its effectiveness. In particular, when the number of skipped layers increases, the performance deviation of random skipping will become much more obvious, *e.g.* ±1.33 for skipping 8 layers. Instead, DAS is still more stable and superior than the random solution, which also suggests the importance of correct redundancy estimations. Overall, these results further confirm the effectiveness and reliability of the proposed DAS.

**Inference Efficiency.** We further compare the actual inference efficiencies of DAS and other methods. The computation overhead during both the training and testing stages are reported in Tab. 3. We can first observe that the PETL methods, *i.e.* LoRA, Adapter and Scaled PA, significantly reduce the computational burden during the training stage. However, during test, these methods lose their efficiency advantage. For instance, Scaled PA is −5.04% slower compared to the full tuning method. In contrast, the proposed DAS enhances the efficiency in both phases. Specifically, DAS only takes similar computation overhead to the PETL methods in the training stage and improves inference efficiency by +19.23%. Overall, these results well confirm the effectiveness of the proposed DAS in the PCETL task.

---

[5]The detailed results are given in our Appendix.

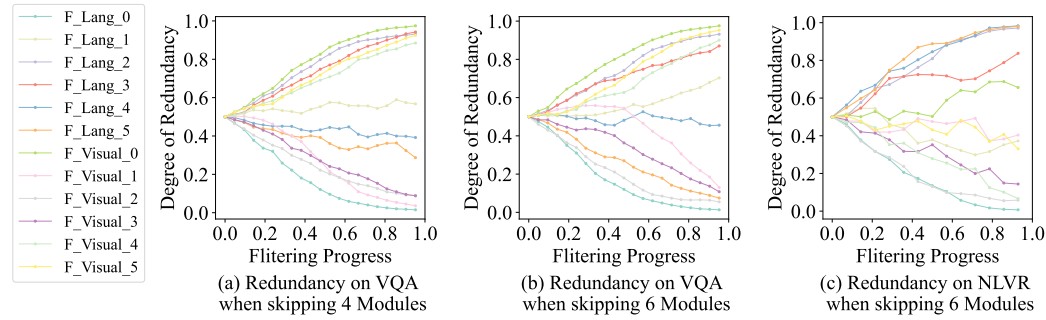

(a) Redundancy on VQA when skipping 4 Modules

(b) Redundancy on VQA when skipping 6 Modules

(c) Redundancy on NLVR when skipping 6 Modules

Figure 4: The change of the redundancies of METER's fusion layers. The horizontal axis shows the progress of training and the vertical axis represents the degree of redundancy based on Eq. 5. F_Lang and F_Visual denote the language and visual modules in the fusion branch.

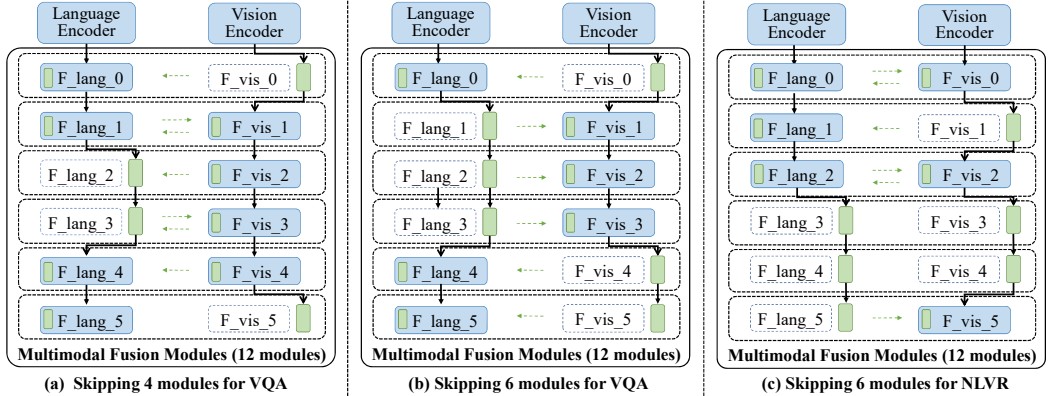

(a) Skipping 4 modules for VQA

(b) Skipping 6 modules for VQA

(c) Skipping 6 modules for NLVR

Figure 5: The optimal subnetworks of METER searched by DAS. Here, the green modules are trainable adapters while the blue ones are frozen during adaption.

### 5.3.2 Qualitative analysis

To obtain deep insight into DAS, we also visualize its changes of redundancy estimations during search, as depicted in Fig. 4. From these plots, we can observe several patterns of DAS across tasks. The first is that the most redundant layers can quickly emerge during DAS, especially when the number of skipped layers is smaller, *e.g.* Fig. 4.a. Meanwhile, for some uncertain layers, their redundancies can be gradually determined after a short period of oscillation. We can also see that the numbers of the skipped language and visual layers are similar on VQA and NLVR. However, the preference about the skipping layers is still different on the two tasks, see Fig. 4.b and Fig. 4.c.

In Fig. 5, we visualize the subnetworks of METER searched by DAS, which are also the final results of Fig. 4. As discussed above, the preferences of the hidden layers of METER are still different for two tasks. In terms of VQA, DAS tends to discard the language modules at the middle level while skipping the visual ones on the top and bottom of the network. In contrast, the language layers skipped on NLVR are all the top ones. These search results can somewhat reflect the properties of these two tasks. For instance, VQA is a reasoning task, thus it focuses more on the high-level semantics of the input text, while NLVR needs a detailed comparison between two images and one sentence, so the last few language layers may be more redundant to this task.

Overall, Fig. 4 and Fig. 5 not only confirm the feasibility of DAS towards PCETL, but also yields some interesting findings about the downstream adaption of VLP models.

## 6 Limitation

Currently, DAS has two main limitations. First, it still needs to set the number of layers to skip. In our future work, we will introduce the computation or hardware constraints to DAS for the more

automatic network skipping. Second, DAS only regards the complete Transformer layers in VLP models as the skipping candidates, limiting its potential in pathway routing. In the future, we will extend the search space with more detailed components, such as MHA and FFN.

## 7 Conclusion

In this paper, we propose a new problem for vision-language pre-trained (VLP) models termed *parameter and computation efficient transfer learning* (PCETL). Existing transfer learning solutions for VLP models can only save the parameter expenditure during downstream task adaption, *e.g.*, the PETL ones, while the excessive computation is still a unconquered problem. To this end, we propose a novel approach called *Dynamic Architecture Skipping* (DAS) towards effective PCETL. DAS can observe the redundancies of VLP modules to downstream tasks via a reinforcement learning based process, and then skip the redundant ones to speed up inference. Meanwhile, DAS also adopts lightweight adapters to serve the hidden feature adaptions and the short-cut connections, thereby reducing the scale of trainable parameters. On two VLP models and three VL tasks, DAS not only shows a great superiority in reducing computation, but is also on par with the PETL methods in terms of parameter overhead and performance.

## 8 Acknowledge

This work was supported by National Key R&D Program of China (No.2022ZD0118201), the National Science Fund for Distinguished Young Scholars (No.62025603), the National Natural Science Foundation of China (No. U21B2037, No. U22B2051, No. 62176222, No. 62176223, No. 62176226, No. 62072386, No. 62072387, No. 62072389, No. 62002305 and No. 62272401), the Natural Science Foundation of Fujian Province of China (No.2021J01002, No.2022J06001) and the China Fundamental Research Funds for the Central Universities (Grant No. 20720220068).

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

# Parameter and Computation Efficient Transfer Learning for Vision-Language Pre-trained Models

**Qiong Wu**[12], **Wei Yu**[12], **Yiyi Zhou**[12], **Shubin Huang**[1], **Xiaoshuai Sun**[12], **Rongrong Ji**[12*]

[1] Key Laboratory of Multimedia Trusted Perception and Efficient Computing,
Ministry of Education of China, Xiamen University, 361005, P.R. China.
[2] Institute of Artificial Intelligence, Xiamen University, 361005, P.R. China.
{qiong, weiyu}@stu.xmu.edu.cn, zhouyiyi@xmu.edu.cn,
shubinhuang@stu.xmu.edu.cn, {xssun, rrji}@xmu.edu.cn

## A  The detailed skipping results

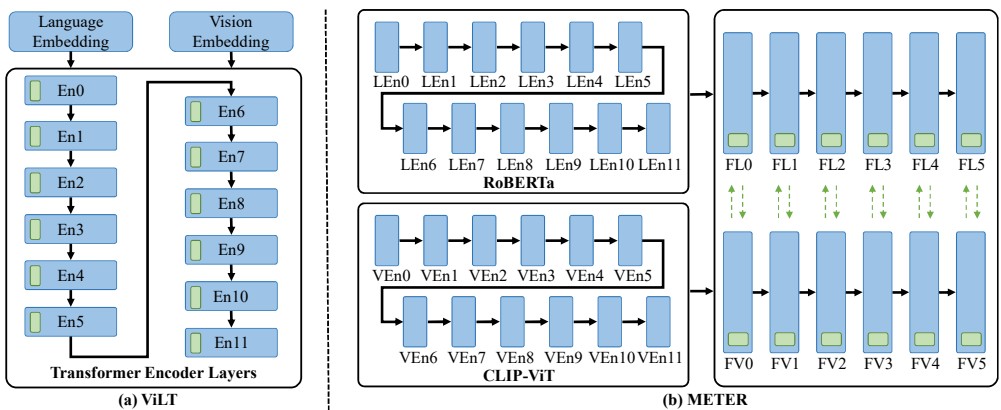

Figure 1: Architectures of the baseline models (a) ViLT and (b) METER. The blue modules are the default Transformer layers that are frozen during the adaptation, while the green ones are the trainable adapters. "En" denotes the encoding layers. "LEn" and "FEn" represent the encoding layers of METER for texts and images, and "FL" and "FV" are the fusion layers for language and vision, respectively.

The architectures of two based models are given in Fig. 1. We also report their detailed skipping results by DAS in Tab. A. Here, "LEn" represents Language Encoder, and "VEn" represents Vision Encoder. We can first see that ViLT is a relatively compact model to METER, which only has 12 Transformer layers without any modality-specific encoder. In this case, it can only be skipped one or two layers without obviously degrading the performance. In stark contrast, METER is a deep and huge VLP model, of which redundancy is much more obvious. By skipping up to 8 layers, its performance drops are still marginal on all tasks. Meanwhile, we also observe that discarding its visual encoder layers will greatly disturb its training and performance during experiments, thus these layers are not considered as the skipping candidates. From Tab. A, we also have some interesting observations. For instance, the language encoding layers are less important to VQA. This may suggest that most questions in VQA2.0 are shorter and less complex, and the model needs to focus more on the visual understanding and cross-modal interactions. This case is less significant on NLVR[2], which requires a detailed comparison between images and texts. Overall, these results confirm that the large VLP models exhibit obvious redundancy to downstream VL tasks. More importantly, the importance of their modules is different to different tasks, requiring proper estimations.

---

*Corresponding Author.

37th Conference on Neural Information Processing Systems (NeurIPS 2023).

Table A: The kipped layers and performance for different base models and tasks. For VQA, we report the test-Dev as the performance. For NLVR$^2$, we report the test-P as the performance. For Flickr30k, we report IR/TR R@1 as the performance. "Fusion" refers to only skipping the layers in the multimodal fusion modules of METER, while "Global" denotes the skipping scope of the fusion modules and the language encoder.

| METER | | | | | |
|---|---|---|---|---|---|
| **Datasets** | **Candidates** | **Number of Skipped** | **Per.** | **Additional FLOPs** | **Skipped Layers** |
| | - | 0 | 75.28 | 1.68G | - |
| | Fusion | 2 | 74.92 | -9.06G | FV0, FV4 |
| | | 4 | 74.80 | -11.16G | FL2, FL3, FV0, FV5 |
| | | 6 | 74.67 | -17.58G | FL1, FL2, FL3, FV0, FV4, FV5 |
| VQA | | 8 | 73.70 | -24.00G | FL1, FL2, FL3, FL4, FV0, FV1, FV4, FV5 |
| | Global | 2 | 75.24 | -3.96G | FV0, LEn6 |
| | | 4 | 75.13 | -4.51G | FV0, LEn10, LEn11 |
| | | 6 | 75.02 | -5.06G | FV0, LEn4, LEn6, LEn8, LEn10, LEn11 |
| | | 8 | 74.05 | -5.61G | FV4, LEn4, LEn5, LEn6, LEn8, LEn9, LEn10, LEn11 |
| | - | 0 | 81.28 | 0.99G | - |
| | Fusion | 2 | 80.07 | -2.66G | FL4, FV5 |
| | | 4 | 80.11 | -4.14G | FL2, FL3, FL5, FV5 |
| | | 6 | 78.16 | -9.97G | FL3, FL4, FL5, FV1, FV3, FV4 |
| NLVR$^2$ | | 8 | 79.30 | -11.45G | FL1, FL2, FL3, FL4, FL5, FV0, FV3, FV4 |
| | Global | 2 | 81.37 | -2.19G | FV5, LEn1 |
| | | 4 | 81.34 | -3.67G | FL2, FL3, FV5, LEn4 |
| | | 6 | 80.04 | -4.22G | FL2, FL5, FL6, LEn5, LEn6, LEn11 |
| | | 8 | 79.61 | -8.34G | FL2, FL3, FL4, FL5, FV1, FV5, LEn4, LEn11 |
| | - | 0 | 81.20/92.40 | 1.68G | - |
| Flickr30k | Fusion | 4 | 80.12/91.80 | -11.16G | FL4, FL5, FV0, FV3 |
| | Global | 4 | 80.42/91.40 | -6.06G | FL2, FL5, FV0, LEn8 |
| ViLT | | | | | |
| **Datasets** | **Candidates** | **Number of Skipped** | **Per.** | **Additional FLOPs** | **Skipped Layers** |
| | - | 0 | 70.13 | 0.73G | - |
| VQA | Global | 1 | 69.28 | -1.03G | En3 |
| | | 2 | 67.64 | -2.79G | En1, En3 |
| | - | 0 | 76.26 | 0.73G | - |
| NLVR$^2$ | Global | 1 | 74.89 | -1.03G | En5 |
| | | 2 | 73.00 | -2.79G | En5, En11 |
| Flickr30k | - | 0 | 62.44/82.10 | 0.73G | - |
| | Global | 1 | 60.66/80.80 | -1.03G | En7 |

## B  The results of random sampling

Table B: The detailed experiment results of random sampled subnetworks for Fig.3 in the main paper.

| METER | | | | | |
|---|---|---|---|---|---|
| **Datasets** | **Candidates** | **Number of Skipped** | **VQA test-Dev** | **Additional FLOPs** | **Skipped Layers** |
| | | | 74.24 | -11.16G | FL2, FL4, FV2, FV3 |
| | | 4 | 74.67 | -11.16G | FL1, FL5, FV0, FV3 |
| | | | 74.08 | -11.16G | FL1, FL4, FV1, FV4 |
| | | | 74.05 | -17.58G | FL1, FL2, FL3, FV2, FV3, FV5 |
| VQA | Fusion | 6 | 73.26 | -17.58G | FL0, FL1, FL4, FV0, FV2, FV5 |
| | | | 73.03 | -17.58G | FL2, FL4, FL5, FV1, FV2, FV5 |
| | | | 71.81 | -24.00G | FL0, FL1, FL3, FL5, FV2, FV3, FV4, FV5 |
| | | 8 | 68.56 | -24.00G | FL1, FL3, FL4, FL5, FV1, FV2, FV3, FV4 |
| | | | 69.88 | -24.00G | FL0, FL2, FL5, FV1, FV2, FV3, FV4, FV5 |

Tab. B gives the detailed results of random sampling mentioned in Fig.3 of the main paper. We can see that random sampling is not only consistently worse than our DAS, but also varies greatly in terms of skipped layers and performance, especially when the number of skipped layers is large. On the contrary, these results just confirm the effectiveness of the proposed DAS.

## C  Generalization on Pre-trained Language Model

To validate the generalization ability of DAS, we also apply it to a pre-trained language model called RoBERTa [5], as shown in Tab. C. Due to the time limit, we do not conduct careful tunings for

Table C: Comparison between DAS and PETL methods for RoBERTa on MNLI and SST2. "En" denotes the encoding layers. "Acc." denotes the accuracy.

| Methods | Updated Parameter | Additional FLOPs | MNLI | | SST2 | |
|---|---|---|---|---|---|---|
| | | | Acc. | Skipped Layers | Acc. | Skipped Layers |
| Full Tuning | 124.65M | 0.0 | 87.6 | - | 94.6 | |
| Bit-Fit [6] | 0.10M | 0.0 | 84.7 | - | 93.7 | |
| Pre-fix [4] | 0.14M | 1.20G | 86.3 | - | 94.0 | |
| LoRA [3] | 0.59M | 0.0 | 87.2 | - | 94.2 | |
| Adapter [2] | 0.63M | 0.33G | 87.2 | - | 94.2 | |
| MAM [1] | 0.61M | 0.79G | 87.4 | - | 94.2 | |
| $DAS_1$ | 0.63M | -3.71G | 86.8 | En10 | 94.1 | En10 |
| $DAS_2$ | 0.63M | -7.74G | 86.7 | En10, En11 | 93.9 | En8, En10 |
| $DAS_3$ | 0.63M | -11.77G | 86.2 | En9, En10, En11 | 93.8 | En7, En8, En10 |

RoBERTa. The settings of DAS follow the main paper, while the rest are the same with MAM [1]. From this table, we can first see that DAS is also applicable to pre-trained language models. It can also achieve the target of PCETL in terms of computation and update parameter scales, while obtaining limited performance drops. However, we can also see that the competitiveness of DAS to other PETL methods is slightly worse on MNLI, of which objective is close to the pre-training ones. We think that the task gap may be a potential factor affecting PCETL. Overall, these results well validate the generalization ability of DAS on LLMs towards PCETL.

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
