# OpenReview forum: "Parameter and Computation Efficient Transfer Learning for Vision-Language Pre-trained Models"
_NeurIPS.cc/2023/Conference — NeurIPS 2023 poster_

### Official Review · Reviewer_fJhT · 2023-06-25

**Soundness:** 2 fair
**Presentation:** 2 fair
**Contribution:** 2 fair
**Rating:** 4
**Confidence:** 4

**Summary:**

This paper aims to achieve both parameters and computation efficiency for the transfer learning of pre-trained VLMs. They combine the previous ladder-side tuning adapter with the proposed dynamic architecture skipping technique to achieve such a goal.

**Strengths:**

It is a good idea to achieve both parameters and computation efficiency for the transfer learning of VLM.

**Weaknesses:**

Although it is a good idea to achieve both parameters and computation efficiency for the transfer learning of VLM, the author does not show the pros and cons of the proposed methods clearly.

1. The training cost of the DAS and the benefit of the DAS is not clear. In Table 1, the authors do not show the running time in practice. FLOPs are not equal to the inference speed, especially the reduction of FLIOPs is marginal. Plus, please provide the training cost of DAS.

2. Why only take experiments on classification tasks? The authors claim they propose a new problem, PCETL, but lack experiments on import VL tasks - image captioning. Is the parameter pruning of VLM not suitable for the image captioning tasks? Does the layer dropping break the text generation ability of pre-trained VLM?

3. Why only provide retrieval results on Flickr30K, it is small and easy. A large model may be unnecessary on Flickr30K.  Please provide retrieval results on MSCOCO.

4. How does the proposed DAS compare to the pruning method in EfficientVLM[1] or other layer pruning methods in the pruning area? Will DAS be necessary? A random baseline in Figure 3 is not enough.


[1] https://arxiv.org/pdf/2210.07795.pdf




**Questions:**

1. Why the method is called "Dynamic Architecture Skipping"? Where the dynamic comes from? From my point of view, the skipped modules are fixed after training and it will not change with the input during inference.

**Limitations:**

As I discuss in the Weaknesses, the authors do not fully address the limitations of the proposed mehtods.

---

> ### Author Rebuttal · Authors · 2023-08-09
>
> # Response to Reviewer #fJhT
> We highly appreciate your time and effort in reviewing this paper, and your valuable feedback has been instrumental in improving the paper. Below, we response to your key concerns point by point.
>
> **Comment 1:** Although it is a good idea to achieve both parameters and computation efficiency for the transfer learning of VLM, the author does not show the pros and cons of the proposed methods clearly. The training cost of the DAS and the benefit of the DAS is not clear. In Table 1, the authors do not show the running time in practice. FLOPs are not equal to the inference speed, especially the reduction of FLIOPs is marginal. Plus, please provide the training cost of DAS.
>
> **Response:** Thanks for this constructive comment. Following your suggestion, we report these results in the following table.
>
> Table A: Comparison of DAS and PETL methods on efficiency for METER.
>
> |Method|VQA Test-Dev|Training Memory (GB)|Training Time|Inference Memory (GB)|Inference Speed (Sample/s)|
> |-|-|-|-|-|-|
> |Full Tuning|77.43|\>40G|N/A|6.8|133.27 |
> |LoRA|74.00|21.5|27h|6.8|133.27 (+0.00%)|
> |Adapter | 74.70 | 22.9 | 28h | 7.2 | 130.93 (-1.75%) |
> |Scaled PA | 75.11 | 23.1 | 30h | 7.1 | 126.50 (-5.08%) |
> |DAS4-Global | 75.09 | 21.7 (search) / 18.1 (training) | 10h (search) + 20h (training) | 6.5 | 146.34 (+9.81%) |
> |DAS4-Fusion | 74.80 | 21.7 (search) / 20.6 (training) | 10h (search) + 18h (training) | 6.5 | 158.79 (+19.14%) |
>
> It can be first seen that our training expenditure is comparable to most PETL methods. Our memory overhead is similar with LoRA during layer searching, and it is also slightly reduced during training. Meanwhile, the search process is quick, and the training hours are also fewer than the PETL methods since DAS has fewer adapters to train. In this case, the overall training expenditure is indeed not expensive.
>
> The inference efficiency is also notable. The memory saving is about 4.4%, while the inference speeds up to 19.14\%.  Similar improvements can be also seen on LLaMA-7B, while the performance is even better.
>
> **Comment 2:** Why only take experiments on classification tasks? The authors claim they propose a new problem, PCETL, but lack experiments on import VL tasks - image captioning. Is the parameter pruning of VLM not suitable for the image captioning tasks? Does the layer dropping break the text generation ability of pre-trained VLM?
>
> **Response:** Thanks for this constructive comment. Following your suggestion, we apply our DAS to LLaMA and report its results on ScienceQA [a], a generative QA benchmark. Here, we follow the settings of LaVIN [c], which can be regarded as our baseline.
>
> Table B: Comparison of DAS and PETL methods on ScienceQA for LLaMA.
>
> | Method | Updated Parameters | FLOPs | Modality Natural | Modality Social | Modality Language | Context  Text | Context Image | Context No | Grade G1-6 | Grade G7-12 |  Avg  |
> |-|-|-|-|-|-|-|-|-|-|-|-|
> | LLaVA-13B  | 13B | - | 90.36 | 95.95 | 88.00 | 89.49 | 88.00 | 90.66 | 90.93 | 90.90 | 90.92 |
> | LaVIN-7B | 3.8M | 833 | 89.25 | 94.94 | 85.24 | 88.51 | 87.46 | 88.08 | 90.16 | 88.07 | 89.41 |
> | DAS4-7B | 44.26M | 729 (-18.61%) | 90.54 | 94.26 | 86.82 | 89.74 | 87.65 | 89.76 | 90.97 | 89.26 | 90.36 |
> | DAS6-7B | 44.26M | 678 (-24.85%) | 89.96 | 94.71 | 87.18 | 89.00 | 87.7 | 89.97 | 90.75 | 89.32 | 90.24 |
>
> It can be seen that our DAS can not only greatly reduces the FLOPs and even improve the performance by skipping 6 layers of LLaMA, which is indeed significant. These results also validate the generalization of DAS on text generation tasks.
>
> **Comment 3:** Why only provide retrieval results on Flickr30K, it is small and easy. A large model may be unnecessary on Flickr30K. Please provide retrieval results on MSCOCO.
>
> **Response:** Thanks for this suggestion. Following your suggestion, we report the retrieval results on MSCOCO in the following table, where the target of PCETL can be still met on this benchmark.
>
> Table C: Comparison of DAS and Finetuning methods on COCO retrieval for METER.
>
> |Method|Update Parameter | Additional FLOPs | COCO IR@1 | COCO IR@5 | COCO IR@10 | COCO TR@1 | COCO TR@5 | COCO TR@10 |
> |-|-|-|-|-|-|-|-|-|
> | Full Tuning | 323.31M | 0.0 | 54.85 | 81.41 | 89.31 | 72.96 | 92.02 | 96.26 |
> | DAS4-Fusion | 5.34M | -9.54% | 54.22| 79.36 | 87.67 | 71.56 | 91.17 | 94.79 |
> | DAS4-Global | 6.23M | -8.68% | 54.60| 80.36 | 88.42 | 72.06 | 91.42 | 95.42 |
>
> **Comment 4**: How does the proposed DAS compare to the pruning method in EfficientVLM[1] or other layer pruning methods in the pruning area? Will DAS be necessary? A random baseline in Figure 3 is not enough.
>
> **Response:** Thanks for recommending this excellent work. However, pruning methods like EfficientVLM are not applicable to PCETL. On one hand, most pruning methods require another full tuning on the downstream tasks, against the target of PCETL. On the other hand, existing PETL methods like Adapter are unable to be combined with the pruning methods, since pruning methods often skip/prune parameter-wise components. We will cite and discuss EfficientVLM in our new version.
>
> **Comment 5:** Why the method is called "Dynamic Architecture Skipping"? Where the dynamic comes from? From my point of view, the skipped modules are fixed after training and it will not change with the input during inference.
>
> **Response:** Thanks for this question. Large pre-trained models like LLaMA are often transferred to various down-stream tasks for practical use. In this case, we think that our DAS can provide optimal inference paths for each task during its real-world applications.
>
> **Reference**
>
> [1] EfficientVLM: Fast and Accurate Vision-Language Models via Knowledge Distillation and Modal-adaptive Pruning.
>
> [a] Learn to Explain: Multimodal Reasoning via Thought Chains for Science Question Answering.
>
> [b] LLaMA: Open and Efficient Foundation Language Models.
>
> [c] Cheap and Quick: Efficient Vision-Language Instruction Tuning for Large Language Models.

---

> > ### Comment · Reviewer_fJhT · 2023-08-14
> > **Respone to the rebuttal**
> >
> > Thanks for your response.
> >
> > - **Comment 1** Does the inference speed mean the model can process 100+ samples in one second on VQA Test-Dev?
> >
> > - **Comment 2** It is surprising to see the method can also achieve good performance after dropping 6 layers. However, there are still a few questions.
> >     - Why the #updated parameters is more than LaVIN-7B? What is the performance of the PEFT baseline with roughly the same #updated parameters?
> >     - The Mem and Time should also be reported with LLaMA.
> >     - So, METER + DAS will not work for image captioning?
> >
> > - **Comment 3** The performance of DAS on COCO is not as good as the performance of DAS on Flickr30k. So, there is a relationship between model size, the difficulty of the task, and the pruning choice.
> >
> > - **Comment 4** We can consider the work as pruning + Adapter. Not many parameters have been pruned (compared to previous pruning methods), so the performance can be recovered by tuning the adapter. A big problem here is that the paper does not discuss and compare previous pruning methods thoroughly. Why do we need Dynamic Architecture Skipping? Why we cannot use some similar pruning methods like [1,2,3]? I do not see any discussion of previous pruning methods in the paper. Personally, I think this is not respectful of the previous pruning works.
> >
> > - **Comment 5** The authors are talking about the generalization of the proposed methods. What does "dynamic" mean? Why use the word dynamic?
> >
> > [1] https://aclanthology.org/2021.emnlp-main.829.pdf
> >
> > [2] https://arxiv.org/pdf/2111.15127.pdf
> >
> > [3] https://proceedings.mlr.press/v202/shi23e/shi23e.pdf

---

> > > ### Author Response · Authors · 2023-08-15
> > >
> > > # Comment to Reviewer #fJhT
> > > Many thanks for your reply.  We hope that our following responses can further address your concerns.
> > >
> > > **Response to Comment 1:** Thanks for this comment. It is tested with a batch size of 32 on one A100. For on-line inference, the speed is about 4.96 samples per second (4.96 (DAS) v.s. 4.16 (Full Tuning)). For a better clarity, we will replace it with on-line inference in our final version.
> > >
> > > **Response to Comment 2.1:** Thanks for your insightful question. The main reason is that DAS not only serves to feature adaption, i.e., PETL, but also to connect the skipped layers. Since LLaMA is a giant model with much larger feature dimensions, its skipped layers require a larger Adapter to connect.  In contrast, the PETL method LaVIN still follows the low-rank property [a], so increasing updated parameters is in fact counterproductive, see the table below.
> > >
> > > | Method | Updated Parameters|FLOPs|Modality Natural|Modality Social|Modality Language|Context Text|Context Image|Context No|Grade G1-6|Grade G7-12|Avg|
> > > | -| -|-|-|-|-|-|-|-|-|-|-|
> > > |LaVIN-7B|3.8M|833|89.25|94.94|85.24|88.51|87.46|88.08|90.16|88.07|89.41|
> > > |LaVIN-7B|44.26M| 838| 84.37 | 74.35 | 86.27 | 82.70 | 73.48 | 89.55 | 83.33 | 81.74 | 82.76 |
> > > |DAS6-7B|44.26M|678 (-24.85%)|89.96|94.71|87.18|89.00|87.7|89.97|90.75|89.32|90.24|
> > >
> > > In fact, the number of updated parameters is still small for LLaMA, only takes about 0.63%.
> > >
> > > **Response to Comment 2.2:** Thanks for your suggestion, and the expenditure of DAS on LLaMA is given below.
> > >
> > > | Method | Avg | Training Memory (GB) | Traning Time | Inference Memory (GB) | Inference Speed (sample/s, batch size=64) | FLOPs (G) |
> > > |-|-|-|-|-|-|-|
> > > |LaVIN-7B | 89.41 | 35 | 6h |40.1|3.51|833|
> > > |DAS4-7B | 90.36 | 36 (search) / 33 (training) | 1h (search) + 4h (training) | 36.5|	3.88 (+10.54%)|	729 (-18.61%)|
> > > |DAS6-7B | 90.24 | 36 (search) / 32 (training) | 1h (search) + 4h (training) | 34.7 |	4.09 (+16.52%)|	678 (-24.85%)|
> > >
> > > **Response to Comment 2.3:** Due to the time limit, we only report the new results of LLaMA on ScienceQA, which is also a generative task and can better validates our generalization. Follow your suggestion, we report the results of BLIP+DAS for image captioning in the following table since METER cannot be directly applied to this task. Note that, these experiments are directly conducted without careful tuning.
> > >
> > > |Method|Update Parameter|FLOPs (G)|Bleu@4|CIDEr|
> > > |-|-|-|-|-|
> > > |Full Tuning|223.97M|100.00%|39.4|131.4|
> > > |DAS4|5.37M|67.49%|37.8 (95.93%)|124.5 (94.74%)|
> > >
> > > It can be seen that DAS can reach about 96 % BLEU performance of full tuning, while saving about 97.6\% updated parameters and up to 33.17% FLOPs. These results are consistent with the target of PCETL, and we think that they can be better with more experimental trials.
> > >
> > > **Response to Comment 3:** In fact, the performance on COCO is slightly better than that on Flickr 30k. For instance, the performance of DAS-4 is about 99.1% of full tuning on COCO, while it is about 97.34% on Flickr30k.
> > >
> > > **Response to Comment 4:** Thanks for your detailed comment. The main difference between DAS and your mentioned works [1,2,3] is that DAS not only needs to reduce the redundant computation,  but also to consider the parameter efficiency, which are the targets of PCETL.
> > >
> > > We agree that existing pruning methods can greatly reduce the parameter size of the target model, but most of them [1,2,3]  require another full tuning on the downstream tasks, which is against the target of PCETL.
> > >
> > > In contrast, DAS can effectively skip the redundant layers and connect the remaining ones with adapters, thereby achieving both of the above goals at the same time. In this case, we think that the contributions of existing pruning methods and our DAS are orthogonal.
> > >
> > > More importantly, the other contribution of this paper is the propose of a new transfer learning task for large-scale pre-trained models, i.e., Parameter and Computation Efficient Transfer Learning (PCETL), which is of great significance to the community and highly recognized by other reviewers.
> > >
> > > Following your suggestion, we will add more discussions about the existing pruning methods to our final version.
> > >
> > > **Response to Comment 5:** As discussed in the previous comment, Large pre-trained models like LLaMA are often transferred to various down-stream tasks. During their practical use, we can add a task notification and apply DAS to dynamically change the routing paths of the model for the inputs of different tasks. In this case, we term this method as the ``dynamic'' one.
> > >
> > > **Reference**
> > >
> > > [a] Edward J. Hu, Yelong Shen, Phillip Wallis, *et al*; LoRA: Low-Rank Adaptation of Large Language Models.
> > >
> > > [1] Francois Lagunas, Ella Charlaix, Victor Sanh, *et al*; Block Pruning For Faster Transformers.
> > >
> > > [2] Hao Yu, Jianxin Wu; A unified pruning framework for vision transformers.
> > >
> > > [3] Dachuan Shi, Chaofan Tao, Ying Jin, *et al*; UPop: Unified and Progressive Pruning for Compressing Vision-Language Transformers.

---

> > > > ### Comment · Reviewer_fJhT · 2023-08-15
> > > > **Response to Comment to Reviewer #fJhT**
> > > >
> > > > Thanks for your response, again.
> > > >
> > > > - **Comment 1** Thanks for your clarification.
> > > >
> > > > - **Comment 2.1** It is weird to see [LaVIN-7B 44.26M] is worse than [LaVIN-7B 3.8M].
> > > >
> > > > - **Comment 2.2** Thanks for the evaluation.
> > > >
> > > > - **Comment 2.3** I think DAS is not good at image captioning on COCO.
> > > >
> > > > - **Comment 3** I re-check the METER paper [1]. In Table 8, 82.22 is the **zero-shot** results. Do the searching and training of DAS use the Flickr or COCO data? I do not find the detail about the training data of DAS in the paper.
> > > >
> > > > - **Comment 4** Well, the main technique contribution is the DAS pruning methods. My main concern is that this part may be replaced by previous pruning methods. The authors do not demonstrate the necessity of the proposed DAS.
> > > >
> > > > - **Comment 5** I re-check Algorithm 1. I have a few questions.
> > > >     - $\theta_K$ is fixed after the algorithm. It will not change in a certain downstream task. If my understanding is not wrong, I still cannot understand why DAS is Dynamic.
> > > >     - Algorithm 1 needs $D_t, D_v$, what are the validation sets in different tasks? Will the use of the validation set be unfair for other PEFT methods?
> > > >
> > > > [1] https://arxiv.org/pdf/2111.02387.pdf

---

> > > > > ### Author Response · Authors · 2023-08-15
> > > > >
> > > > > # Comment to Reviewer #fJhT
> > > > > Thanks again for your replies.
> > > > >
> > > > > **Response for Comment 2.1:** Thanks for this comment. A main hypothesis behind existing PETL methods is that the weight change in an over-parameterized model has ``a low intrinsic rank'' during their fine-tune, so that PETL methods can use lightweight and low-rank modules to approximate this change [1]. In this case, continually increasing the number of updated parameters does not guarantee performance gains. Similar results can be also seen on LoRA[2].
> > > > >
> > > > > [1] A. Aghajanyan, L. Zettlemoyer, S. Gupta; Intrinsic dimensionality explains the effectiveness of language model finetuning.
> > > > >
> > > > > [2] Zi-Yi Dou, Yichong Xu, Zhe Gan, *et al*; An Empirical Study of Training End-to-End Vision-andLanguage Transformers.
> > > > >
> > > > > **Response for Comment 2.3:** We agree that the performance gain on COCO captioning is slightly less than other tasks, but its effect is still significant considering the parameter and training expenditures. Meanwhile, this result still has a large room to improve, since we only perform one-round trial in this rebuttal.
> > > > >
> > > > > **Response for Comment 3:** In fact, the retrieval result of 82.22 is reported in Tab.9, which is obtained under the fine-tuning setting, see the caption of Tab.9 in METER. The training setting of DAS is the same with METER, and we will publicly release our source code to facilitate the reproduction.
> > > > >
> > > > > **Response for Comment 4:** Thanks again for this question. As we responded in Reviewer #LdaL, PECTL is a new task that unexplored, and we find it difficult for existing solutions to directly achieve the goals of PCETL simultaneously.  In this case, we propose a new approach, i.e., DAS, and proves that PCETL is a feasible yet important task for large-scale pre-trained models. We think that this is the necessity of DAS.
> > > > >
> > > > > We acknowledge that there exist solutions with similar concepts to DAS, such as dynamic networks [1,2] and the pruning methods [3,4,5] you mentioned, but most of them have obvious shortcomings towards PCETL. For instance, we quantitatively compared some of these alternatives in our rebuttal, such as MRM [3] and J2C [6], of which results are much inferior to our DAS and hard to meet PCETL.
> > > > >
> > > > > In terms of pruning methods, most of them require full tuning on downstream tasks, which is against PCETL as discussed before. Meanwhile, for the methods pruning weight neurons, they are also intractable to combine with adapters, which usually serves the transformation for feature vectors or tensors. Undoubtedly, there are potential ways to make existing pruning methods applicable for PCETL, but we think that this is not the focus of this paper.
> > > > >
> > > > > [1] Dinghuai Zhang, Kartik Ahuja, Yilun Xu, *et al*; Can Subnetwork Structure be the Key to Out-of-Distribution Generalization?
> > > > >
> > > > > [2] Chaitanya Devaguptapu, Samarth Sinha, K. J. Joseph, *et al*; Δ-Networks for Efficient Model Patching.
> > > > >
> > > > > [3] Francois Lagunas, Ella Charlaix, Victor Sanh, *et al*; Block Pruning For Faster Transformers.
> > > > >
> > > > > [4] Hao Yu, Jianxin Wu; A unified pruning framework for vision transformers.
> > > > >
> > > > > [5] Dachuan Shi, Chaofan Tao, Ying Jin, *et al*; UPop: Unified and Progressive Pruning for Compressing Vision-Language Transformers.
> > > > >
> > > > > [6] Alexander Yom Din, Taelin Karidi, Leshem Choshen; Jump to Conclusions: Short-Cutting Transformers With Linear Transformations.
> > > > >
> > > > > **Response for comment 5.2:** Thanks for this comment. In our experiments, all the compared methods and our DAS share the same settings of these base VL models, including the validation splits. We will describe the experimental settings in more detail to avoid misunderstanding, and our source codes will be also released.

---

> > > > > > ### Comment · Reviewer_fJhT · 2023-08-15
> > > > > > **Response to Comment to Reviewer #fJhT**
> > > > > >
> > > > > > Thanks for your response.
> > > > > >
> > > > > > - **Comment 4** DAS combines pruning + PEFT. So, DAS should be better than PEFT in efficiency and better than pruning in performance. This is the meaning of the so-called PECTL task. There are many works that aim to achieve retraining free pruning [1] and block-level pruning rather than weight or filter-level pruning. My concern is that DAS does not justify itself by comparing or discussing such works.
> > > > > >
> > > > > >
> > > > > > [1] https://openreview.net/pdf?id=0GRBKLBjJE

---

> > > > > > > ### Author Response · Authors · 2023-08-15
> > > > > > >
> > > > > > > # Comment to Reviewer #fJhT
> > > > > > > Thanks again for your reply.
> > > > > > >
> > > > > > > **Response for Comment 4:** Thanks again for your detailed comment.
> > > > > > >
> > > > > > > In fact, the retraining free method your mentioned still requires the model to be fully tuned for the downstream task. In other words, whether retraining free or not, the existing pruning methods still needs to store a large number of updated parameters on the downstream task, e.g., at least 50% in [1], while most PETL methods and our DAS keep this ratio within 1%. Thus, we can see that to obtain both computation and parameter efficiencies simultaneously is the key challenge of PCETL, rather than accomplishing them individually.
> > > > > > >
> > > > > > > In terms of network pruning, we agree that block-level pruning is likely to achieve PCETL, similar to the previously mentioned dynamic network approaches. We also welcome more practitioners can extend them to PCETL based on the code project of this paper.
> > > > > > >
> > > > > > > As said before, we will follow your suggestion to add more discussions about network pruning in our final version, including the ones we made during rebuttal.
> > > > > > >
> > > > > > > Lastly, thanks again for your time and effort in reviewing this paper, and your valuable feedbacks have been instrumental in improving our work.
> > > > > > >
> > > > > > > [1] Woosuk Kwon, Sehoon Kim, Michael W. Mahoney, *et al*; A Fast Post-Training Pruning Framework for Transformers.

---

> > > > > > > > ### Comment · Reviewer_fJhT · 2023-08-18
> > > > > > > > **Final question**
> > > > > > > >
> > > > > > > > Thanks for the response from the authors.
> > > > > > > >
> > > > > > > > One thing still bothers me, could the authors make a clear clarification on the validation set $D_v$ in Alg. 1? For example, what is $D_v$ for retrieval on Flickr30k and COCO?
> > > > > > > >
> > > > > > > > I would like to give a summary of my opinion about the paper and end the discussions in this thread after this question.

---

> > > > > > > > > ### Author Response · Authors · 2023-08-18
> > > > > > > > >
> > > > > > > > > Thanks again for your reply.
> > > > > > > > >
> > > > > > > > > In Algorthim 1, $D_v$ is either the default validation set or the excluded splits of the training data of these benchmarks. In particular, the validation sets of Flickr30k and COCO are the ones defined in [1]. For VQA2.0, $D_v$ is the default validation split. In terms of NLVR2, $D_v$ is an excluded subset from the training data. We will add these details to our final version.
> > > > > > > > >
> > > > > > > > > Your time and effort are both highly appreciated.
> > > > > > > > >
> > > > > > > > > [1] Karpathy A., Fei-Fei L; Deep visual-semantic alignments for generating image descriptions.

---

> > > > > > > > > > ### Comment · Reviewer_fJhT · 2023-08-18
> > > > > > > > > > **Summary of the discussion**
> > > > > > > > > >
> > > > > > > > > > Thank the authors for their immediate response.
> > > > > > > > > >
> > > > > > > > > > I would like to make a summary and end the conversation.
> > > > > > > > > >
> > > > > > > > > > - (**main concern**) I still hold my opinion that the paper lacks thorough analysis and comparison with previous pruning methods to justify the necessity of the proposed DAS. From my point of view, it is easy to incorporate previous pruning methods + PEFT methods by setting the parameters in the backbone as unlearnable.
> > > > > > > > > >
> > > > > > > > > > - I cannot understand why DAS is dynamic as $\theta_K$ is unchanged during inference.
> > > > > > > > > >
> > > > > > > > > > - The usage of the validation set in experiments needs more clarification. For example, whether the PEFT methods also use the validation set.
> > > > > > > > > >
> > > > > > > > > >
> > > > > > > > > > I still hold my original rating, borderline reject. However, I will not advocate for a rejection during the next reviewer discussion period.

---

### Official Review · Reviewer_5vAx · 2023-07-05

**Soundness:** 4 excellent
**Presentation:** 4 excellent
**Contribution:** 3 good
**Rating:** 7
**Confidence:** 3

**Summary:**

The paper proposes a dynamic architecture skipping (DAS) method for the parameter and computation efficient transfer learning (PCETL) problem. DAS explores the optimal short-cut pathway in VLP models. Extensive experiments show the effectiveness of DAS both in reducing computation and parameters.

**Strengths:**

The paper introduces a novel and intriguing approach by considering network jumps as k-armed bandit sampling. It highlights the significance of reducing computational complexity in Visual Language Pretraining (VLP) models, providing strong motivation for the proposed methodology. The paper is well-written and effectively communicates its ideas. Experiments are solid, with necessary analysis and ablations for different parts of the method.

**Weaknesses:**

Compared to the number of parameters and FLOPS, it would be better to add the training time and GPU memory cost as metrics.

**Questions:**

N/A. Already written in the weakness section.

**Limitations:**

Yes.

---

> ### Author Rebuttal · Authors · 2023-08-09
>
> # Response to Reviewer 5vAx
> We highly appreciate your time and effort in reviewing this paper. Your comments and feedback are instrument to the improvement of our work.
>
> **Comment 1:** Compared to the number of parameters and FLOPS, it would be better to add the training time and GPU memory cost as metrics.
>
> **Response:** Thanks for this comment. Following your suggestion, we report the training time and GPU memory cost in the following table.
>
> | Method | VQA Test-Dev | Training Memory (GB) | Training Time |
> |-|-|-|-|
> | Full Tuning | 77.43 | \>40G | N/A |
> | LoRA | 74.00 | 21.5 | 27h |
> | Adapter | 74.70 | 22.9 | 28h |
> | Scaled PA | 75.11 | 23.1 | 30h |
> | DAS4-Global | 75.09 | 21.7 (search) / 18.1 (training) | 10h (search) + 20h (training) |
> | DAS4-Fusion | 74.80 | 21.7 (search) / 20.6 (training) | 10h (search) + 18h (training) |
>
> It can be seen that our memory overhead is similar with LoRA during layer searching, and it can be further slightly reduced during training. Meanwhile, the search process is quick, and the training hours are also fewer than the PETL methods since DAS has fewer adapters to train. In this case, the overall training expenditure is not significantly more expensive than the PETL methods.

---

> > ### Comment · Reviewer_5vAx · 2023-08-18
> >
> > Thank you for your response. From the perspective of training memory and time, It seems that DAS does not have a great advantage. However, this method is still worthy of recognition in terms of novelty.

---

> > > ### Author Response · Authors · 2023-08-18
> > >
> > > Many thanks for your reply, and your valuable feedbacks have been instrumental in improving this paper.

---

### Official Review · Reviewer_pBnL · 2023-07-07

**Soundness:** 3 good
**Presentation:** 2 fair
**Contribution:** 2 fair
**Rating:** 5
**Confidence:** 4

**Summary:**

The paper presents dynamic architecture skipping (DAS), which can drop some transformer layers during inference. The routing of DAS is learned by reinforcement learning. To make the training parameter efficient, every layer has an adapter for training. After training, DAS drops several redundant layers and replaces them with adapters to reduce the inference FLOPs.

**Strengths:**

1. Most parameter-efficient training methods add extra cost in inference, and it is interesting to explore how to reduce the cost.
2. The design of the proposed approach, which replaces some transformer layers of the backbone model with adapters, is reasonable.

**Weaknesses:**

1. Several baselines [2] and related works [1, 2] are missing. I think the paper would be stronger if it can compare to [2].
2. I am not sure what is the uniqueness of the approach only applies to the VL domain. If the approach is general, I would expect applying the approach to some other models and tasks too (LLM or ViT).

[1] Fan, Angela, Edouard Grave and Armand Joulin. “Reducing Transformer Depth on Demand with Structured Dropout.”

[2] Din, Alexander Yom, Taelin Karidi, Leshem Choshen and Mor Geva. “Jump to Conclusions: Short-Cutting Transformers With Linear Transformations."

**Questions:**

1. The training pipeline is more complex than the other approaches since the training involves search, redundancy observation, and final fine-tuning. I wonder how much cost (e.g. training time and memory) is needed to train the method compared to other approaches?
2. Are the results on one run or multiple runs? If it is on one run, I would suggest using the average of multiple runs to justify the robustness of the approach.
3. Given the GFLOPs saving, how much the approach can improve the inference speed and inference memory?

---
**Post-rebuttal**
Thank you for the authors' response. I have read it and it addressed my questions.

**Limitations:**

The limitation in the paper is sufficient.

---

> ### Author Rebuttal · Authors · 2023-08-09
>
> # Response to Reviewer #pBnL
> We highly appreciate your time and effort in reviewing this paper, as well as your encouraging and constructive comments on our work. Below, we response to your key concerns point by point.
>
> **Comment 1:** Several baselines [2] and related works [1, 2] are missing. I think the paper would be stronger if it can compare to [2].
>
> **Response:**  Thanks for this suggestion. We will cite and discuss your mentioned excellent works. Following your suggestion, we supplement the comparison with J2C [2] in the following table, which is also combined with adapters like our DAS.
>
> Table A: The comparison between DAS and the suggested baseline for METER.
>
> | Method|Updated Parameter|VQA test-dev|VQA Additional FLOPs|NLVR2 test-P|NLVR2  Additional FLOPs |
> | -|-|-|-|-|- |
> | Full Tuning|323.31M|77.43|0%|83.05|0.00|
> | Classifier Only|-|69.93|0%|73.23|0.00 |
> | J2C-2|4.18M|67.34|-11.52%|68.89|-8.83% |
> | J2C-4|3.58M|69.26|-18.60%|69.08|-15.95% |
> | DAS4-Fusion|5.34M|74.80|-11.97%|80.11|-9.70% |
> | DAS6-Fusion|5.34M|75.67|-18.86%|79.30|-17.72% |
>
> It can be seen that in the absence of layer redundancy evaluation, the effect of layer skipping by J2C is not satisfactory, and its performance greatly lags behind our DAS.
>
> **Comment 2:** I am not sure what is the uniqueness of the approach only applies to the VL domain. If the approach is general, I would expect applying the approach to some other models and tasks too (LLM or ViT).
>
> **Response:** Thanks for this constructive suggestion. We have applied our DAS to LLaMA-7B [a] on ScienceQA [b] following the settings of LaVIN [c]. The results are given in Table. B.
>
> Table B: Comparison of DAS and PETL methods on ScienceQA for LLaMA.
>
> | Method|Update Params|FLOPs(G)|Modality Natural|Modality Social|Modality Language|Context Text|Context Image|Context No|Grade G1-6|Grade G7-12|Avg |
> |-|-|-|-|-|-|-|-|-|-|-|-|
> | LLaVA-13B|13B|-|90.36|95.95|88.00|89.49|88.00|90.66|90.93|90.90|90.92 |
> | LaVIN-7B|3.8M|833|89.25| 94.94| 85.24| 88.51| 87.46| 88.08| 90.16| 88.07| 89.41 |
> | DAS4-7B| 44.26M|729 (-18.61%)|90.54|94.26|86.82|89.74|87.65|89.76|90.97|89.26|90.36 |
> | DAS6-7B|44.26M|678 (-24.85%)|89.96|94.71|87.18|89.00|87.7|89.97|90.75|89.32|90.24 |
>
> By saving up to 25% FLOPs, our DAS can even achieve better results than LaVIN on LLaMA-7B. Notably, its best performance is very close to LLaVA[d] with LLaMA-13B while saving much more expenditure. These results well confirm our DAS towards the target of PCETL.
>
> **Comment 3:** The training pipeline is more complex than the other approaches since the training involves search, redundancy observation, and final fine-tuning. I wonder how much cost (e.g. training time and memory) is needed to train the method compared to other approaches?
>
> **Response:** Thanks for this comment. Following your suggestion, we report the training costs of DAS and the compared methods in the following table.
>
> Table C: Comparison of DAS and PETL methods on efficiency for METER.
>
> | Method|VQA test-dev|Training Memory (GB)|Training Time |
> |-|-|-|-|
> | Full Tuning|77.43|\>40|N/A |
> | LoRA|74.00|21.5|27h |
> | Adapter|74.70|22.9|28h |
> | Scaled PA|75.11|23.1|30h |
> | DAS4-Global|75.09|21.7 (search) / 20.6 (training)|10h (search) + 20h (training) |
>
> It can be seen that our memory overhead is similar to LoRA during layer searching, and it can be further slightly reduced during training. Meanwhile, the search process is quick, and the training hours are also fewer than the PETL methods since DAS has fewer adapters to train. In this case, the overall training expenditure is not significantly more expensive than the PETL methods.
>
> **Comment 4:** Are the results on one run or multiple runs? If it is on one run, I would suggest using the average of multiple runs to justify the robustness of the approach.
>
> **Response:** Thanks for this suggestion. In fact, the search result of our DAS is very stable. On METER, we test DAS with three random seeds, and the skipped layers are the same. In this case, we only report the training result on one run, similar to the PETL methods. Following your suggestion, we will provide the multiple-run ones in our new version.
>
> **Comment 5:** Given the GFLOPs saving, how much the approach can improve the inference speed and inference memory?
>
> **Response:**  Thanks for this question. The detailed improvements are given in the following tables. In terms of METER, the memory saving is about 4.4%, while the inference speeds up to 19.14\%.  Similar improvements can be also seen on LLaMA-7B, while the performance is even better.
>
> Table D: Comparison of DAS and PETL methods on inference efficiency for METER.
>
> | Method|VQA Test-Dev|Inference Memory (GB)|Inference Speed(Sample/s)|FLOPs |
> |-|-|-|-|-|
> | Full Tuning|77.43|6.8|133.27|93.2 |
> | LoRA|74.00|6.8|133.27 (+0.00%)|93.2 (-0.0%) |
> | Adapter|74.70| 7.2|130.93 (-1.75%)|94.9 (+1.82%) |
> | Scaled PA|75.11|7.1|126.50 (-5.08%)|94.3 (+1.18%) |
> | DAS4-Global|75.09|6.5|146.34 (+9.81%)|88.7 (-4.82%) |
> | DAS4-Fusion|74.80|6.4|158.79 (+19.14%)|82.1 (-11.9%) |
>
> Table E: Comparison of DAS and PETL methods on inference efficiency for LLaMA-7B.
>
> | Method|Avg|Inference Memory (GB)|Inference Speed (sample/s)|FLOPs (G) |
> |-|-|-|-|-|
> | LaVIN-7B|89.41|40.1|3.51|833 |
> | DAS4-7B|90.36|36.5|3.88 (+10.54%)|729 (-18.61%) |
> | DAS6-7B|90.24|34.7|4.09 (+16.52%)|678 (-24.85%) |
>
> **Reference**
>
> [1] Reducing Transformer Depth on Demand with Structured Dropout.
>
> [2] Jump to Conclusions: Short-Cutting Transformers With Linear Transformations.
>
> [a] LLaMA: Open and Efficient Foundation Language Models.
>
> [b] Learn to Explain: Multimodal Reasoning via Thought Chains for Science Question Answering.
>
> [c] Cheap and Quick: Efficient Vision-Language Instruction Tuning for Large Language Models.
>
> [d] Visual Instruction Tuning.

---

### Official Review · Reviewer_LDaL · 2023-07-10

**Soundness:** 3 good
**Presentation:** 3 good
**Contribution:** 3 good
**Rating:** 6
**Confidence:** 4

**Summary:**

This work focuses on the problem of transfer learning in the context of vision-language pre-trained (VLP) models. Existing works that adapt VLP models primarily address the issue of parameter efficient transfer learning (PETL). However, these methods do not effectively reduce the computation complexity of VLP models. Therefore, this work takes PETL a step further and introduces a new setting called parameter and computation efficient transfer learning (PCETL). The goal of PCETL is to reduce the computation complexity of pre-training models, enabling faster inference, while only tuning a fraction of parameters.

To achieve PCETL, a method called dynamic architecture skipping (DAS) is proposed. DAS assists in finding the optimal subnetwork routing of VLP models for downstream tasks. Although the concept of dynamic architecture skipping has been extensively explored in transfer learning for image spaces, it has not yet been investigated in the era of large-scale pre-trained models. The proposed approach demonstrates promising results on two VLP models and three vision-language benchmarks."

**Strengths:**

1. The paper is well-written and easy to follow.
2. The concept of dynamic skipping, as yet unexplored in the era of large-scale VLP models, holds considerable promise. Its introduction to VLP models could significantly reduce computational and time complexity.
3. As the daily release of pre-trained large-scale models continues to increase, methods for efficiently transferring learned features become increasingly significant.
4. The experimental results compellingly illustrate the potential of the proposed methodology.

**Weaknesses:**

1. The concept of dynamic architecture skipping is not new and is already well-established in the case of transfer learning for standard imagery (see references below). It would be beneficial if the authors could provide reasons why existing approaches are unsuitable for the current problem and setting.
2. The proposed approach is not end-to-end differentiable and models the problem as a k-armed bandit. A more comprehensive end-to-end approach would be ideal (refer to references 2 and 4).
3.  The paper lacks a comparison with existing dynamic architecture skipping methods, which is a crucial element for an encompassing evaluation.
4. The authors acknowledge that the number of layers to be skipped must be manually determined. Some of the transfer learning methodologies listed below are capable of handling this automatically, borrowing ideas from there can benefit this work.


**References**
1. [BlockDrop: Dynamic Inference Paths in Residual Networks](https://arxiv.org/abs/1711.08393)
2. [SpotTune: Transfer Learning through Adaptive Fine-tuning](https://arxiv.org/abs/1811.08737)
3. [Can Subnetwork Structure be the Key to Out-of-Distribution  Generalization?](https://arxiv.org/abs/2106.02890)
4. [$\Delta$-Networks for Efficient Model Patching](https://arxiv.org/abs/2303.14772)

**Questions:**

1. Looks like random skipping is a very strong baseline for this work (Line 256-257). The performance difference between random skipping and the proposed approach appears marginal, yet considering that random skipping is less costly, are these gains still substantial?
2. Could you please explain the rationale for modeling this as a k-armed bandit problem? Additionally, what would be the primary challenges in solving this problem in an end-to-end differentiable manner?

**Limitations:**

Limitations Section discusses the limitations of this work clearly

---

> ### Author Rebuttal · Authors · 2023-08-09
>
> # Response to Reviewer #LdaL
> We highly appreciate your time and effort in reviewing this paper, as well as your encouraging and constructive comments on our work. Below, we response to your key concerns point by point.
>
> **Comment 1:** The concept of dynamic architecture skipping is not new and is already well-established in the case of transfer learning for standard imagery (see references below).
>
> **Response:** Thanks for this suggestion. In this paper, we propose a new task called Parameter and Computation Efficient Transfer Learning (PCETL) for large-scale pre-trained models, which should not only reduces the computation redundancy but also needs to avoid the expensive full fine-tune on various downstream tasks.
>
> This is a new field yet unexplored, and we find it difficult for existing solutions to directly achieve the above goals simultaneously. Thus, we propose a novel approach called Dynamic Architecture Skipping based on the k-armed bandit theory.
>
> In terms of your mentioned methods, most of them have obvious shortcomings towards PCETL. BlockDrop is an example-dependent method that uses a policy network to predict the routing path for each input. Its combination with PETL methods like Adapter is intractable, since the problem definition as well as the training scheme should be largely changed. SpotTune is an approach for choosing which layers to finetune rather than to skip. Δ-networks is unable to reduce the computation since its weighted connections only serve to adaption.
>
> A potential solution is MRM, which learns gating functions to decide layering options. In fact, its computation reduction is very non-deterministic, and in some cases, it does not skip redundant layers, see Table A.
>
> **Comment 2:**  A more comprehensive end-to-end approach would be ideal (refer to references 2 and 4).
>
> **Response:** Thanks for this comment. Following your suggestion, we compare our DAS with MRM [3] and Δ-Networks [4] in the following table.
>
> Table. A: The comparison between DAS and alternative methods.
>
> | Method|Updated Parameter|Training Memory (GB)|Training Time|VQA  test-dev|VQA  Additional FLOPs|NLVR2 test-P|NLVR2 Additional FLOPs |
> | -|-|-|-|-|-|-|- |
> | Full Tuning|323.31M|\>40|N/A|77.43|0.0|83.05|0.00 |
> | Classifier Only|\-|21.4|27h|69.93|0.0|73.23|0.00 |
> | MRM(1e-5)|6.23M|26.3 (search) /24.0 (training)|8h (search) +22h (training)|75.28|+1.80%|81.39|-4.16% |
> | Δ-Networks|36|21.5|27h|67.34|+0.00%|71.49|+0.00% |
> | DAS2-Global|6.23M|21.6 (search) /20.5 (training)|11h (search) +22h (training)|75.24|-4.25%|81.37|-4.16% |
> | DAS4-Global|6.23M|21.7 (search) /20.6 (training)|10h (search) +20h (training)|75.09|-4.84%|80.69|-6.95% |
>
> We can first observe that the adaption of Δ-Networks is much inferior to our DAS, and it also cannot reduce the computation as we mentioned above.
> MRM can be extended to a PETL method when combined with Adapters.
> However, its computational efficiency is unstable and difficult to directly constrain on different VL tasks. Simply put, MRM is likely not to choose to skip layers during training. More importantly, its GPU memory overhead is much more expensive than our method.
>
> Overall, our DAS is still the best choice considering efficiency and effectiveness.
>
> **Comment 3:** The paper lacks a comparison with existing dynamic architecture skipping methods.
>
> **Response:**  Following your suggestion, we supplement the comparison with MRM in Tab. A, from which we can see that our DAS is the best trade-off between efficiency and performance.
>
> **Comment 4:** The authors acknowledge that the number of layers to be skipped must be manually determined. Some of the transfer learning methodologies listed below are capable of handling this automatically, borrowing ideas from there can benefit this work.
>
> **Response:** Thanks for this insightful comment. In fact, the manual setup of DAS is an advantage over the automatic solutions.
>
> To explain, different from Network Architecture Search (NAS), the pre-trained VL models have a fixed structure, which greatly limits the choice of network skipping. In this case, gradient-based methods can only rely on the training loss to automatically select which layer should be skipped, and this process is unstable and uncontrollable. As shown in Tab. A, MRM keeps all Transformer layers of METER on VQA2.0, but our method can reduce up to four layers with similar performance.
>
> Overall, our DAS can better meet the PCETL requirement in practice.
>
> **Comment 5:** Looks like random skipping is a very strong baseline for this work (Line 256-257)?
>
> **Response:**  In fact, the number described in Line 256-257 refers to the performance deviation of random skipping. The actual performance gains of our method are obvious, e.g. +5.6\% when skipping 8 layers, as shown in Fig. 3.
>
> **Comment 6:** Could you please explain the rationale for modeling this as a K-armed bandit problem? Additionally, what would be the primary challenges in solving this problem in an end-to-end differentiable manner?
>
> **Response:** Thanks for this question. As discussed above, in PCETL, the structures of VLP models are fixed, so we cannot change the network depth to reduce the computation just like NAS. In this case, we model network skipping as a k-armed bandit problem, *i.e.*, which k layers can be skipped, and evaluate the policy via numerous single-shot samplings. This setting can directly set the computation cost for PCETL.
>
> In contrast, in PCETL, the gradient-based or differentiable methods like MRM are uncontrollable for layer skipping, as shown in Tab. A, since its search results are attributed to the training loss rather than the pre-defined target.
>
> **References**
>
> [1] BlockDrop: Dynamic Inference Paths in Residual Networks.
>
> [2] SpotTune: Transfer Learning through Adaptive Fine-tuning.
>
> [3] Can Subnetwork Structure be the Key to Out-of-Distribution Generalization?
>
> [4] Δ-Networks for Efficient Model Patching.

---

> > ### Comment · Reviewer_LDaL · 2023-08-21
> >
> > I have carefully reviewed the initial submission and the authors' response. I appreciate the effort that has been invested in addressing the concerns raised, and I would like to thank the authors for the new results. The responses provide answerers to my questions, and in light of this, I have updated my rating accordingly.

---

### Author Rebuttal · Authors · 2023-08-09

We highly appreciate AC for pushing forward NeurIPs 2023, and also thank all reviewers for their valuable and encouraging comments on this paper, such as \`\`*Its introduction to VLP models could significantly reduce computational and time complexity.*\'\' by Reviewer LDaL, \`\`*The experimental results compellingly illustrate the potential of the proposed methodology.*\'\' by Reviewer LDaL, \`\`*The design of the proposed approach is reasonable.*\'\' by Reviewer pBnL, \`\`*introduces a novel and intriguing approach*\'\' by Reviewer 5vAx , \`\`*It highlights the significance of reducing computational complexity in Visual Language Pretraining (VLP) models*\'\' by Reviewer 5vAx, *et al*.

During the rebuttal phrase, our main responses include:

1. The details of training costs, including GPU memory and training time, are similar to most PETL methods.

2. The actual inference speed-up, which can be +19% in practice.

3. The comparison with alternative skipping methods, where our merits can be still witnessed.

4. The application to LLaMA on ScienceQA, where the target of PCETL is still achieved by our methods.

Meanwhile, the key concerns of all reviews are also point-by-point responded in each rebuttal.

Here, we would like to emphasize our key contributions again:

1. We raise a new problem called Parameter and Computation Efficient Transfer Learning (PCETL) for VLP models.

2. We propose a novel Dynamic Architecture Skipping (DAS) for PCETL, which can greatly reduce the computation redundancy on downstream tasks.

Lastly, the new results in rebuttal will be added to our final version, and our source codes will be publicly released after acceptance.

Best,

The authors.

---

### Decision · Program_Chairs · 2023-09-21

**Decision:**

Accept (poster)

**Comment:**

This paper proposes a PETL (parameter-efficient transfer learning) method by utilizing dynamic architecture skipping. It receives 3 positive reviews and 1 slightly negative reviews. The raised issues include similarity upon existing DAS methods, application range, and experimental  validation. In the rebuttal phase, the authors provide sufficient explanation and results to convince 3 reviewers remain positive. While the 4th reviewer is slightly negative,  the remaining issues are minor from the PETL perspective. Overall, the AC has checked all the files, and welcomes this designing reporting in this venue. The authors shall incorporate the additional results and analysis in the camera-ready version.